# Clinical and Topographic Screening for Scoliosis in Children Participating in Routine Sports: A Prevalence and Accuracy Study in a Spanish Population

**DOI:** 10.3390/jcm14010273

**Published:** 2025-01-06

**Authors:** José María González-Ruiz, Nada Mohamed, Mostafa Hassan, Kyla Fald, Eva de los Ríos Ruiz, Pablo Pérez Cabello, Álvaro Rubio Redondo, Bruna da Rosa, Thomaz Nogueira Burke, Lindsey Westover

**Affiliations:** 1Department of Mechanical Engineering, University of Alberta, Edmonton, AB T6G 2G8, Canada; nada1@ualberta.ca (N.M.); mkhassan@ualberta.ca (M.H.);; 2Society for the Advancement of Applied Computer Science Berlin, GFaI Gesellschaft zur Förderung Angewandter Informatik e. V., Volmerstraße 3, D-12489 Berlin, Germany; 3Independent Researcher; evadelosriosruiz@gmail.com; 4Asociación de Escoliosis de Castilla y León (ADECYL), Arroyo de la Encomienda, 47195 Valladolid, Spain; pablofisio@adecyl.org (P.P.C.);; 5Department of Biomedical Engineering, University of Alberta, Edmonton, AB T6G 2G8, Canada; bdarosa@ualberta.ca (B.d.R.); lwestove@ualberta.ca (L.W.); 6Allied Health Institute, Federal University of Mato Grosso do Sul, Campo Grande 79070-900, Brazil; thomaz.burke@ufms.br

**Keywords:** scoliosis, screening, surface topography, prevalence, sport

## Abstract

**Background**: Idiopathic scoliosis (IS) is a common spinal deformity affecting 0.5% to 5.2% of children worldwide, with a higher reported range in Spain (0.7–7.5%). Early detection through screening is crucial to prevent the progression of mild cases to severe deformities. Clinical methods such as the ADAM test and trunk rotation angle (TRA) are widely used, but the development of three-dimensional (3D) surface topography (ST) technologies has opened new avenues for non-invasive screening. The objectives of this study were (1) to perform clinical and ST-based scoliosis screening in a sample of healthy children involved in club sports, (2) to estimate the agreement between clinical and ST screening methods, (3) to describe the prevalence of scoliosis by sport, sex, and age, and (4) to evaluate the diagnostic performance of both screening approaches using available radiographs as a reference standard. **Methods**: A total of 343 children (58.7% males, 41.3% females; mean age 11.69 ± 2.05 years) were screened using both clinical and ST methods. Clinical screening included the ADAM test and TRA measurement, while ST screening was performed using BackSCNR^®^, a markerless 3D scanning software. The children with positive screening results were recommended to obtain radiographs to confirm the diagnosis. Kappa agreement, sensitivity, specificity, positive predictive value (PPV), negative predictive value (NPV), and accuracy were calculated for both screening modalities using radiographic results as the gold standard. **Results**: The prevalence of scoliosis was 3.2% (n = 11) based on radiographic confirmation. The prevalence by sport was highest in swimming (17.6%), with minimal differences by sex (males 3.6%, females 2.5%). The clinical screening showed a sensitivity of 73%, specificity of 97%, PPV of 47%, NPV of 99%, and accuracy of 96%. The ST screening showed a sensitivity of 36%, specificity of 99%, PPV of 80%, NPV of 97%, and accuracy of 97%. The kappa values indicate a moderate influence of chance for both methods (clinical *κ* = 0.55; ST *κ* = 0.48). The balanced accuracy was 84% for the clinical screening and 68% for the ST screening. **Conclusions**: The clinical screening method showed superior sensitivity and balanced accuracy compared to ST screening. However, ST screening showed higher specificity and PPV, suggesting its potential as a complementary tool to reduce the high positive predictive value. These results highlight the importance of combining screening methods to improve the accuracy of the early detection of IS in physically active children, with the radiographic confirmation of the positive screened cases remaining essential for accurate diagnosis.

## 1. Introduction

Idiopathic scoliosis (IS) is a complex, three-dimensional condition that disrupts the natural symmetry of the trunk in otherwise healthy children [1,2,3]. It is the most common form of spinal deformity, ranging from 0.5% to 5.2% [4]. This condition can have a significant impact on quality of life, particularly in terms of impaired self-image perception and mental health [5]. Moderate to severe cases may also be associated with pain and decreased functional ability [6]. While most cases present as mild (less than 20° Cobb in the coronal plane), IS exhibits an unpredictable and occasionally progressive nature [7]. In recent years, there has been an increase in the prevalence of IS, especially among physically inactive girls [8]. These findings suggest a possible relationship between sedentarism and IS, but, on the other hand, a clear relationship between exercise and IS has not been established. Given the potentially progressive nature of IS, an escalation in prevalence could result in a greater number of children and adolescents requiring bracing for moderate cases or even surgical intervention for the most severe cases. Curiously, a higher prevalence range than the global average has been observed in Spain (0.7–7.5%) [2].

Previous studies have reported that 5% of the cases have evidence of curve progression above 30° [9]. However, there is also heterogeneity in these reports, with 2/3 of the scoliosis cases considered potentially progressive during adolescence [4]. To mitigate this progression, screening tests have become a widespread practice for the early detection of trunk deformities that can be managed conservatively to avoid progression from mild to moderate or severe cases [2,10]. There is evidence that early screening can reduce the rate of surgery by identifying, on average, milder curves that may benefit from conservative management [10,11]. On the other hand, patients diagnosed late usually have deformities greater than 40°, which may exceed the indication for bracing [9]. Thus, both age and curve severity are known to be among the most reliable variables for predicting progression [1].

The most commonly used screening methods consist of the combination of the ADAM test and the trunk rotation angle (TRA), which is assessed using a scoliometer to measure the inclination of the trunk protrusions [2,12,13]. A widely accepted TRA threshold is 5°, with a sensitivity of 94% in identifying curves of 20° or greater. It has a global sensitivity of 71% and a specificity of 83%, making it the most technically efficient test with robust interobserver and intraobserver reliability, especially in thoracic curves [10,14,15]. When the ADAM test and TRA are combined with Moiré topography, sensitivity (88.1–93.8%) and specificity (99.2%) are further increased [2,4,10]. Despite these reported values, a significant number of false positives (0.8–21.5%) occur in clinical screening, particularly in early adolescents and children younger than 11 years [4,13]. This fact, in addition to the costs associated with widespread implementation, is an argument against generalized screening programs [10]. Nevertheless, Tomaru et al. defended that if a single operation could be avoided by early bracing after screening, it might be cost-effective [16].

While earlier screening methods, such as Moiré topography, have been used, their high cost has hindered their widespread use in screening programs. The advent of new 3D screening technologies that rely on markerless asymmetry analysis and use simple, accessible tools, such as an iPad and a structured light sensor, has opened new avenues for clinicians. Surface topography (ST), in comparison to traditional screening, offers advantages regarding objectivity (due to the non-manual measurements) and 3D comprehension of torso asymmetry [17]. In this context, our group has developed a convolutional neural network (CNN) model to classify asymmetry patterns observed in healthy adolescents and those with AIS based on torso surface scans [18]. ST analysis to compute torso asymmetries involved reflecting the 3D geometry of the torso and aligning with the original torso by minimizing the distance between points. Then, the deviations across the dorsal surface between the torsos, as well as in depth, were utilized for training the CNN model. The outputs of the model are probability distributions with outcome bins 0 (healthy) and 1 (AIS). The model has shown some promising initial results in our testing dataset. Specifically, an accuracy of 95%, sensitivity 97%, and specificity 90% were observed. Further details of the model can be found in Mohamed et al. [18]. It has been integrated into BackSCNR^®^, Version 3.2.0 (https://backscnr.com/; “URL (accessed on 31 May 2025 for analysis)”), an iPad-compatible ST software that uses a structure sensor to capture the full torso geometry in the standing position. The unique features of this software are that individuals do not need to perform the forward bending of the ADAM test, and that the model could be prospectively trained and improved with new cases.

Considering all the above, the objectives of the present study were as follows:To perform a clinical screening intervention, based on the ADAM test and TRA, and a ST screening in a sample of healthy children participating in a club-based sports activity routine.To estimate the agreement between the clinical and topographic screening methods.To describe the prevalence according to different sports, sex, and age groups (juveniles or adolescents) using the available radiographs.To measure the sensitivity, specificity, positive predictive value (PPV), negative predictive value (NPV), and accuracy of both approaches using the available radiographs of positive screening cases.

## 2. Methods

Ethical approval was obtained from the University of Alberta Research Ethics Board, code Pro00135870, and conforms to the World Medical Association Code of Ethics (Declaration of Helsinki). For sample size calculation, we used a confidence level of 0.95 and an estimation error of 0.05, considering an expected prevalence of 0.075 according to the literature, with a minimum sample size estimate of n = 107.

A total of 351 children participating in routine weekly sports activities at local sports schools in Torrelavega (Cantabria, Spain) and Arroyo de la Encomienda (Valladolid, Spain) were recruited voluntarily after obtaining informed consent from parents and participants. The inclusion criteria were as follows: any sport practiced regardless of the number of hours per week, both sexes, and age between 7 and 18 years. The age range is consistent with the age of higher risk for developing IS (adolescent growth spurt) and within the range of other screening studies [12,19]. Adolescents with a previous diagnosis of scoliosis or other spinal conditions, or those who did not meet any of the other inclusion criteria were excluded.

After applying the inclusion and exclusion criteria, the sample size was 343 and basic information was collected, including sport practiced, weekly time spent in sport, sex, and age. Anthropometric data including height, weight, and body mass index (BMI) were also collected. For females, the presence of menarche was recorded.

Clinical screening was performed by a trained physiotherapist using a positive ADAM test result and ≥5° TRA as an indication of possible scoliosis. We chose this value because values ≥ 6–7° of TRA have been reported as a reliable criterion for detecting curves greater than 25°, but it exceeds the limit of mild curves that, if detected early, may allow conservative treatment to be initiated as a precaution. Therefore, the standard procedure was to place the scoliometer over the thoracic and lumbar prominences (if present) and register the higher value (Figure 1).

ST screening was then performed using BackSCNR^®^, as shown in Figure 2.

The results of both the clinical and topographic screening were subjected to a kappa agreement analysis. Subsequently, the families of positive cases from the clinical and/or BackSCNR^®^ screening were recommended to obtain a radiograph to confirm the diagnosis. However, not all the participants with a positive screening result chose to receive a confirmation radiograph.

Prevalence was estimated using true positives confirmed by radiography and an estimate of positives screened without radiography. The literature suggests that there is a high rate of false positives [20], so we used the positive predictive value (PPV) of both methods (clinical and ST) to estimate true and false positives in the group of participants with a positive screening result but no radiographic confirmation.

We performed further analyses using true and false positives from the available radiographs. Negative cases concordant with both methods were considered true negatives. The sensitivity, specificity, PPV, negative predictive value (NPV), and accuracy of the two screening modalities were measured: (A) clinical screening and (B) ST screening. Finally, the kappa and balanced accuracy were estimated for both models to mitigate the effect of unbalanced data (high number of true negatives due to lack of radiological confirmation). Balanced accuracy provides better performance for imbalanced datasets where equal attention to all the classes is important.

Later, a visualization of topographical maps was provided for the participants with positive screening according to ST and 4° of TRA (potentially considered liminal cases but lacking X-ray confirmation) and the participants with a negative ST result but radiological confirmation of low-mild scoliosis (below 15° Cobb).

All the statistical analyses were performed in R Studio (version 2024.04.02). *p*-value < 0.05 was considered statistically significant.

## 3. Results

The mean and standard deviation (SD) values of age, height, weight, BMI, sex distribution, and weekly training hours are shown in Table 1 and Figure 3, classified by sport. A total of 48 out of 343 participants regularly practice a second sport in a weekly routine, which in some cases increases the total number of training hours, as shown in Figure 3.

Table 2 shows a 2 × 2 contingency table of the two screening methods used for concordance analysis. Kappa index, standard error (SE), and 95% confidence interval (CI) were calculated for interpretation. The results were as follows: Kappa index (*κ*) = −0.02; SE(*κ*) = 0.04; and CI95%(*κ*) = (−0.11, 0.06). According to these results, there is no agreement between the methods due to a negative kappa index and a CI that includes zero. There were 298 of 343 individuals with agreement and 45 of 343 without agreement.

The prevalence of AIS observed was 11/343 = 3.2%. Confirmed cases of AIS were detected in handball (1), swimming (3), volleyball (1), rugby (4), and judo (2) with prevalences of 2.4% (handball), 17.6% (swimming), 6.7% (volleyball), 7.4% (rugby), and 2.8% (judo), respectively. The prevalence by sex was 3.6% in males and 2.5% in females. By age, the prevalence was 3% in adolescents (10–18 years) and 3.6% in juveniles (7–10 years). The mean (SD) Cobb angle on positive radiographs was 16.59° (5.81°) corresponding to mild scoliosis. Specifically, only one individual had moderate scoliosis, while the other ten had mild scoliosis.

No statistical differences were found between the age, sex, and anthropometry of the participants diagnosed with AIS (n = 11, confirmed by a Cobb angle greater than 10° on radiographs) and the remaining participants (n = 332). These results are presented in Table 3.

Table 4 shows the true and false positive (TP and FP), true and false negative (TN and FN), and total sample including the n = 297 negative cases screened by both methods and n = 21 cases with the available radiographs for testing sensitivity, specificity, PPV, NPV, and accuracy values. Note that the differences in sample size (26 fewer individuals) for the calculations compared to Table 2 are due to the subjects who were screened positive by clinical and/or ST, but did not have a corresponding radiograph to confirm the diagnosis.

With these available radiologic data, the sensitivity, specificity, PPV, NPV, and accuracy values of the two screening modalities were calculated and the results are shown in Table 5. The results of the kappa index show that both methods show a moderate influence of chance (0.41–0.60) despite their high accuracy values. In terms of balanced accuracy, the clinical screening showed a better balance between its positive and negative predictions than the ST screening.

Figure 4 shows the deviation color map (DCM) of a patient with radiographic evidence of scoliosis with a Cobb angle of 31.2° (Figure 4a) and three suspected cases of scoliosis identified by ST but without radiological confirmation (Figure 4b–d). ST analysis using the DCM provides color patches corresponding to the magnitude of the deviation of the torso relative to the reflected [21]. In BackSCNR^®^, the higher the intensity of red, the greater the hump (curve convexity), while the blue areas represent depressions (curve concavity). Thus, these three participants with positive ST show a 3D asymmetry pattern compatible with scoliosis.

Figure 5 shows the DCM of one confirmed non-scoliosis participant (Figure 5a) compared to four participants who were negative on ST (Figure 5b–e) but positive with radiological confirmation of low–mild scoliosis (less than 15° Cobb). As it can be observed, these four participants do not show substantial 3D deformities due to the absence of significant hump/depression areas despite their mild scoliosis diagnosis.

## 4. Discussion

This study aimed to screen for scoliosis in healthy children using clinical and topographic methods; assess their agreement; analyze prevalence by sport, sex, and age; and evaluate the diagnostic performance of both methods.

The global prevalence, including only radiologically confirmed cases, was 3.2%. This result is similar to that reported by Dunn et al. [4] (1.2–3.5%) and significantly higher than those published by Adobor et al. [22] in Norway (0.55%).

However, one of the limitations of this study was the lack of radiographs in all the participants due to ethical reasons, a potential reason for the underestimation of the prevalence in the screened sample. In previous clinical screenings similar to ours, up to 21.5% of the detected cases were false positives [4,14]. We have 47 positive screened cases from the two methods evaluated, but among them, only 21 reported radiographic confirmation/exclusion. Accordingly, 26 participants without radiologic confirmation were exposed to a false positive rate within 47–80% according to the PPV observed in the ST and clinical screening. This estimate could increase the prevalence to 6.7–9.3%, similar to that reported by Granado et al. [23] and lower than the prevalence (16%) reported by Zurita Ortega et al. [24]. According to radiologic evidence (n = 11), we observed a higher prevalence in males (3.6% vs. 2.5%). This is in agreement with a previous screening study in Spain with 2956 children between 8 and 12 years of age, in which 57.6% of the scoliosis cases detected were observed in boys [24]. Regarding the sex distribution of prevalence, it shows an opposite trend compared to the one reported by Negrini et al. for mild cases (56.5% for females and 43.5% for males [25]. In terms of age of detection and severity, we observed a higher prevalence in juveniles than in adolescents (3.6% vs. 3%), and the mean (SD) Cobb angle was 16.59° (5.81°). The goal of any screening method is to detect mild curves, so our results are within the expected range.

An association between sports activity and the onset of scoliosis has been reported previously [25]. Specifically, ballet, rhythmic gymnastics, and swimming have been identified as risk factors for the development of spinal deformities. On the other hand, sports as a means of physical activity are among the recommendations of the SOSORT guidelines, although the evidence has shown that some of them could contribute to increased joint laxity, i.e., gymnastics [26]. Our screening intervention in active children and adolescents included 19 different sports. Swimming (17.6%), rugby (7.4%), and volleyball (6.7%) had above-average prevalences, but we did not find any cases of scoliosis in children who participated in gymnastics. Our findings are consistent with previous studies where a high prevalence of scoliosis was observed in swimmers and volleyball players [27,28]. To our knowledge, this is the first study to report prevalence in rugby players and given the high prevalence observed, further efforts should be made to identify rugby and similar contact sports as a potential risk factor for the development of scoliosis in its practitioners.

Our second objective was to test the agreement between two screening methods, the most widely used clinical approach based on the ADAMS test and TRA and the CNN model based on ST images. No agreement was observed between the two (*κ* = −0.02). We believe that this significant difference between the screening results is due to the assessment posture used. Our CNN screening model was trained with ST images of the participants in a standing position, while the clinical screening was performed in a forward bending position. Thus, the shape features captured and considered by each are different and potentially complementary during the screening process. In addition, the clinical screening is based on an angular measurement of the dorsal hump, while the ST model used deviations and depth maps of the entire back torso. In light of this result, the evaluation of both methods in terms of sensitivity, specificity, PPV, NPV, and accuracy gained relevance.

According to the literature, sensitivity and specificity greater than 70%, with PPV between 30% and 50%, are required to consider a screening method acceptable [4]. By these standards, the clinical screening performed in our study is within the range, while the ST screening model is below the recommended minimum sensitivity. This means that only 36% of the positive cases were detected by the ST-based model, causing an increased risk of false negatives that could result from the delayed initiation of precocity treatment. On the other hand, differences between the methods were observed for PPV. Although the PPV range observed in the clinical screening is within the accepted range, the high false positive rate (53%) is remarkable. This value is even higher than the previously reported false positive rates using the ADAMS test (up to 40%) and ADAMS plus TRA (up to 21.5%) [4,14,24]. This highlights one of the known limitations of clinical screening, the over-referral to radiological examinations, which leads to the unnecessary irradiation of healthy children and adolescents. The low false positive rate of the ST model could be considered a promising way to avoid this problem.

Due to the main limitation of our study, the lack of radiographs in all the participants, all the results discussed above should be considered with caution. For example, among the positive cases detected by ST, there were three cases with a positive ADAMS test and 4° of TRA that did not receive a radiological confirmation, but their topographies show clear signs of potential scoliosis. The DCM has been associated with spinal curve asymmetries and has been used to predict curve severity and progression [29,30]. These, shown in Figure 4 and compared to a confirmed case of moderate scoliosis, have asymmetry patterns compatible with scoliosis due to the presence of humps and depressions. With the correct radiologic conformation of the condition in these participants, the sensitivity of the ST method would be higher than that currently reported in this study.

On the other hand, there are four cases in which ST had a negative result, but there were radiographs confirming the presence of liminal scoliosis (10–15° Cobb). Consequently, the 3D asymmetries of these four cases could be considered within the range of normality with the reported negative ST result. Indeed, these observations could open a discussion on the differences between the radiologic and the topographic approaches. If scoliosis is defined as a 3D deformity with curves greater than 10° in the coronal plane [31], it is necessary to establish a strong relationship between 3D asymmetries and the Cobb angle. In a previous study by González-Ruiz et al. [32], it has been reported that control subjects may have higher 3D asymmetries than some of the scoliosis patients. Thus, it is possible that having mild scoliosis (10–15° Cobb) does not necessarily imply having a 3D deformity, or at least one that is more relevant than in the non-scoliotic population. Further training of the CNN model with mild cases may increase its accuracy in detecting AIS.

Considering all of the above, a combined screening approach could offer some advantages in the early detection of scoliosis. We could propose the use of the ST screening to reduce the high number of false positive results obtained by clinical screening. Thus, in the first step of the screening process, clinical procedures as described in this study and the previous literature would detect potential scoliosis cases with relatively high sensitivity. Subsequently, the screened positive cases would be subjected to ST analysis, which, due to its high PPV, would reduce the final number of screened cases requiring radiologic confirmation of the condition.

## 5. Conclusions

Scoliosis is a common condition in children in Spain, with average prevalence values higher than those observed in other countries. Participation in some sports such as swimming, rugby, and volleyball has been associated with a higher prevalence than the rest of the sports studied. However, we cannot establish a direct relationship between their practice and the onset or development of scoliosis. There is a large consensus on the ability of early screening to prevent the most severe scoliosis cases and the clinical screening method demonstrated superior sensitivity and balanced accuracy compared to ST screening. However, ST screening showed higher specificity and PPV, suggesting its potential as a supplementary tool. These results highlight the importance of combining screening methods to improve the early detection rates of IS in physically active children, with radiographic confirmation remaining essential for accurate diagnosis.

## Figures and Tables

**Figure 1 jcm-14-00273-f001:**
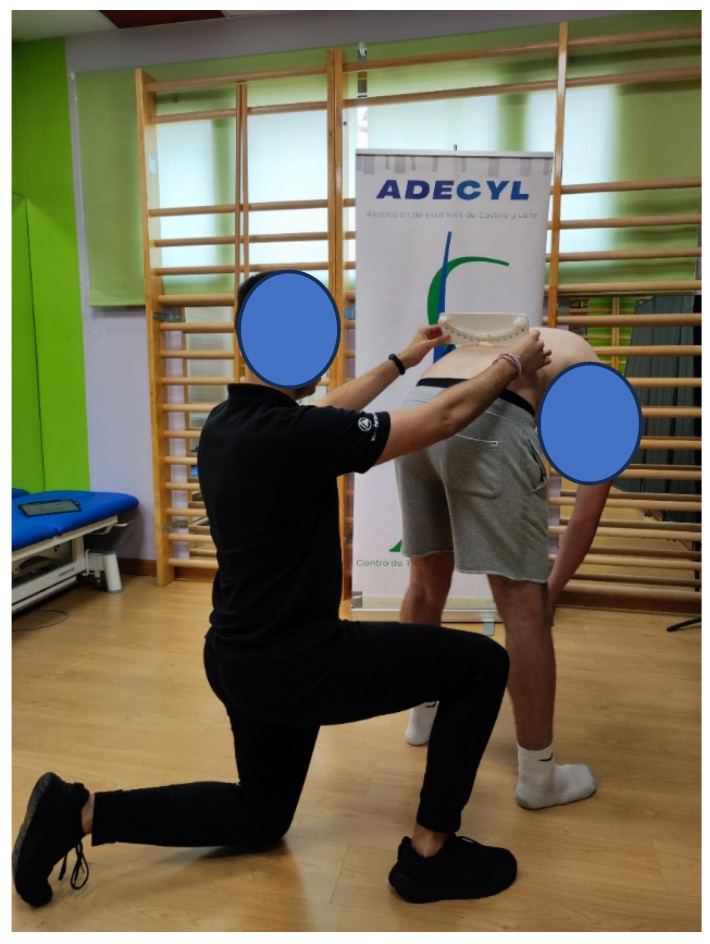
Clinical screening performed by combining the ADAM test and TRA measured with an analog scoliometer.

**Figure 2 jcm-14-00273-f002:**
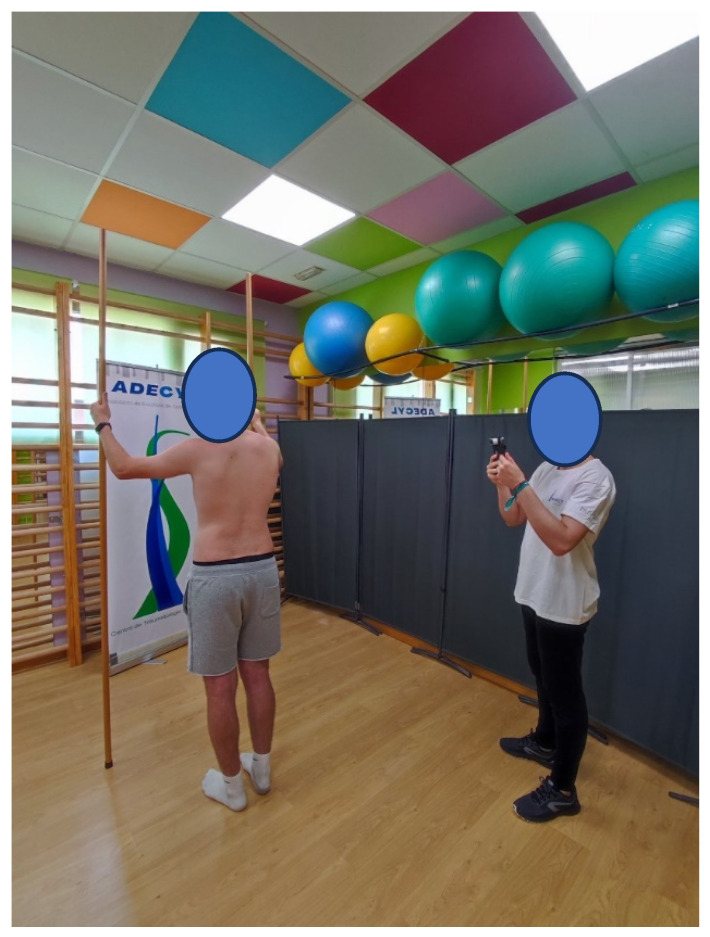
ST screening in standing position. The upper extremities are holding sticks to make the position as comfortable as possible without significantly changing the shape of the trunk.

**Figure 3 jcm-14-00273-f003:**
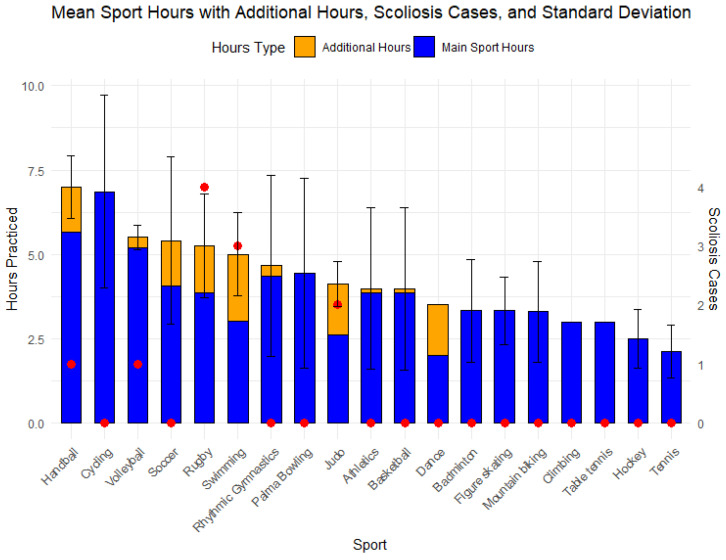
Summary of Weekly Sports Hours (main sport and additional hours). The total hours show in some sports an increment due to some of their practitioners playing a second sport increasing weekly dedication. SD is represented with a black line over the bars. Red dots show scoliosis cases by sport.

**Figure 4 jcm-14-00273-f004:**
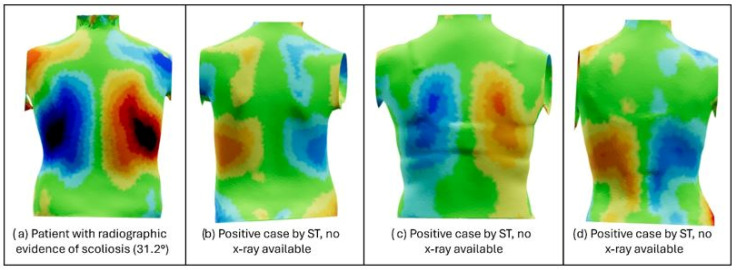
Topographic maps of participants (**b**–**d**) with positive ST screening and nonsignificant TRA (4°) compared to a confirmed case of scoliosis (**a**).

**Figure 5 jcm-14-00273-f005:**
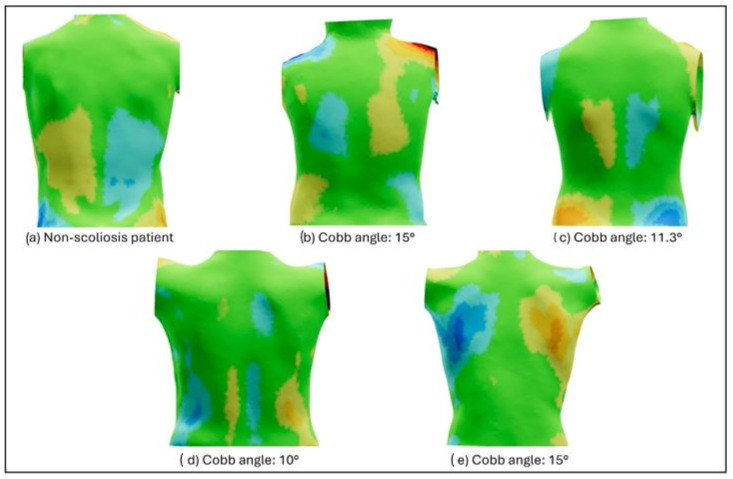
Topographic maps of participants (**b**–**e**) with mild scoliosis and negative ST screening compared to a confirmed case of non-scoliosis (**a**).

**Table 1 jcm-14-00273-t001:** Sample summary. Height in meters and weight in kilograms. Mean and standard deviations (SD) are presented.

Main Sport	N	Males (%)	Females (%)	Mean Age (SD)	Mean Height (SD)	Mean Weight (SD)	Mean BMI (SD)
Athletics	8	62.50	37.50	11.43 (1.29)	1.45 (0.09)	39.99 (10.94)	18.65 (2.91)
Badminton	3	66.67	33.33	13.10 (1.39)	1.63 (0.06)	53.00 (5.46)	20.04 (3.09)
Basketball	12	100.00	0.00	11.76 (2.08)	1.57 (0.16)	47.48 (12.62)	19.00 (2.47)
Climbing	1	0.00	100.00	15.16 (0.00)	1.58 (0.00)	51.90 (0.00)	20.92 (0.00)
Cycling	22	86.36	13.64	12.80 (2.22)	1.55 (0.14)	48.65 (11.11)	19.93 (2.84)
Dance	2	0.00	100.00	9.07 (0.79)	1.35 (0.00)	35.55 (0.49)	19.54 (0.39)
Figure skating	9	0.00	100.00	11.41 (3.98)	1.40 (0.19)	40.23 (19.56)	19.12 (4.35)
Handball	42	85.71	14.29	13.85 (1.76)	1.64 (0.11)	57.46 (14.22)	21.12 (3.79)
Hockey	3	100.00	0.00	10.35 (3.39)	1.39 (0.14)	44.87 (20.72)	22.42 (5.65)
Judo	72	63.89	36.11	9.93 (2.40)	1.40 (0.15)	38.32 (15.01)	18.96 (3.81)
Mountain biking	23	86.96	13.04	10.96 (2.35)	1.45 (0.14)	41.74 (9.28)	19.54 (1.68)
Palma Bowling	9	88.89	11.11	12.18 (3.76)	1.53 (0.19)	51.57 (18.29)	21.36 (3.66)
Rhythmic Gymnastics	30	3.33	96.67	11.71 (2.35)	1.46 (0.13)	40.71 (12.28)	18.71 (3.84)
Rugby	54	83.33	16.67	11.18 (2.93)	1.46 (0.19)	47.41 (19.73)	21.34 (4.70)
Soccer	9	88.89	11.11	10.26 (1.99)	1.40 (0.13)	37.87 (10.50)	19.15 (4.53)
Swimming	17	64.71	35.29	10.36 (1.69)	1.44 (0.12)	37.07 (10.17)	17.69 (2.65)
Table tennis	3	100.00	0.00	10.06 (0.99)	1.38 (0.07)	30.97 (6.05)	16.10 (1.85)
Tennis	9	33.33	66.67	13.52 (1.55)	1.57 (0.10)	51.11 (12.64)	20.57 (3.64)
Volleyball	15	0.00	100.00	12.99 (2.03)	1.57 (0.10)	48.51 (9.13)	19.57 (2.24)
Total	343	58.66	41.34	11.69 (2.05)	1.48 (0.12)	44.44 (11.49)	19.67 (3.06)

**Table 2 jcm-14-00273-t002:** Contingency table of surface topography (ST) and clinical (C) screening results.

	C+	C−	Total
ST+	1	17	18
ST−	28	297	325
Total	29	314	343

**Table 3 jcm-14-00273-t003:** Mean, SD, and statistical test between groups (AIS and no-AIS). Height is expressed in meters and weight in kilograms.

Variable	AIS Mean (SD)	No-AIS Mean (SD)	*t*-Test (*p*-Value)	Chi Square (*p*-Value)
Age	11.91 (2.72)	11.49 (2.69)	0.51 (0.61)	-
Height	1.53 (0.17)	1.48 (0.16)	0.94 (0.34)	-
Weight	48.11 (20.75)	44.69 (15.42)	0.71 (0.47)	-
BMI	19.82 (4.83)	19.77 (3.73)	0.04 (0.96)	-
Males	72.7%	64.5%	-	0.47
Females	27.3%	35.5%		

**Table 4 jcm-14-00273-t004:** Summary of the negative and positive cases that reported an X-ray for diagnosis confirmation allowing for true and false positive (TP and FP) and true and false negative (TN and FN) estimation.

	C	ST
TP	8	4
FP	9	1
TN	297	305
FN	3	7
TOTAL	317	317

**Table 5 jcm-14-00273-t005:** Performance of both screening methods and 95% confidence intervals of sensitivity, specificity, PPV, NPV, and accuracy.

	C	ST
Accuracy (CI 95%)	0.96 (0.94, 0.98)	0.97 (0.95, 0.99)
Kappa	0.55	0.48
Sensitivity	0.73	0.36
Specificity	0.97	0.99
PPV	0.47	0.80
NPV	0.99	0.97
Balanced Accuracy	0.84	0.68

## Data Availability

The raw data supporting the conclusions of this article will be made available by the authors upon request.

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
