# Peer review of "Clinical and Topographic Screening for Scoliosis in Children Participating in Routine Sports: A Prevalence and Accuracy Study in a Spanish Population"

_jcm, 2025, doi:10.3390/jcm14010273_

Round 1
Reviewer 1 Report
Comments and Suggestions for Authors
Clinical and Topographic Screening for Scoliosis in Children Participating in Routine Sports: A Prevalence and Accuracy Study in a Spanish Population.
The objectives of this study were (1) to perform clinical and ST-based scoliosis screening in a sample of healthy children involved in club sports, (2) to estimate the agreement between clinical and ST screening methods, (3) to describe the prevalence of scoliosis by sport, sex, and age, and (4) to evaluate the diagnostic performance of both screening approaches using available radiographs as a reference standard. Using the 343 children sample data, the authors concluded that the importance of combining screening methods to improve the accuracy of early detection of IS in physically active children, with radiographic confirmation of positive screened cases remaining essential for accurate diagnosis.
Critique:
1. Considering the study's aims, the authors focused on healthy children involved in "club sports". However, to draw reasonable research purpose in the Introduction, the authors need to be described the differences between normal population and sports population.
2. Considering the number of cases, I think all IS cases were not presented in this study. All demographics need to be illustrated such as, severity of scoliosis, curve type et al.
3. Minor issues;
Line 121: After applying the inclusion and exclusion criteria, the sample size was n=343 and
- Please re-state for proper expression. (the sample size was 343 or the number of sample was 343)
Figures; I recommend to cover someone face
Author Response
- Considering the study's aims, the authors focused on healthy children involved in "club sports". However, to draw reasonable research purpose in the Introduction, the authors need to be described the differences between normal population and sports population.
Thank you for your comment. Most adolescents do not meet the minimum physical requirements, which may be a reason for the recent increase in the prevalence of IS (based on the prevalence of positive ADAM test in a Croatian population). We have further explained this idea in lines 60-62.
- Considering the number of cases, I think all IS cases were not presented in this study. All demographics need to be illustrated such as, severity of scoliosis, curve type et al.
Unfortunately, this is a limitation of our study, as we did not have access to the images for ethical reasons. As we explain in the Methods section, we suggest to the positive screened families to ask their doctors for radiographs and some of them report back to us the confirmed Cobb. We received radiologic confirmation of 11 cases, giving a prevalence of 3.2%. We haven't presented the cases in detail, especially the curve type, because we only had access to the numerical Cobb angle, not the radiograph. However, thanks to your comment, we have included a severity report on lines 200-201 that shows only 1 moderate case versus 10 mild cases.
Minor issues;
Line 121: After applying the inclusion and exclusion criteria, the sample size was n=343 and
- Please re-state for proper expression. (the sample size was 343 or the number of sample was 343)
Thank you for pointing this out, we have followed your suggestion in line 136.
Figures; I recommend to cover someone face
We covered the heads of both patients, but not the clinicians. They are co-authors of the manuscript and verbally agreed not to cover their faces.

Reviewer 2 Report
Comments and Suggestions for Authors
This is a very well-written study, and I learned a lot from reviewing it. I have a few comments for improvement that I hope will be helpful.
1. In the Introduction, it would be beneficial to provide additional details about ST screening, particularly the technical advantages of 3D topography and how it differs from traditional clinical screening methods. Expanding on these points would help clarify the central role of ST screening in this study. Additionally, briefly highlighting the unique features of the BackSCNR® system could further emphasize its relevance and innovation.”
2. The presentation of Table 2 could be simplified to enhance clarity. Consider converting the data into a bar graph or other visual representation, as this might help readers grasp patterns more intuitively. Alternatively, emphasizing the columns with average values and providing standard deviation data in the footnotes could make the table more concise and reader-friendly.
3. While the high specificity of the ST screening model is promising, the sensitivity of 36% raises significant concerns regarding its clinical utility. The authors should provide a more detailed discussion on the potential implications of this low sensitivity, such as the risk of false negatives leading to delayed treatment for mild scoliosis cases. Additionally, the manuscript would benefit from an exploration of the factors contributing to the low sensitivity, such as limitations in detecting mild asymmetries or issues related to the measurement posture.
Author Response
In the Introduction, it would be beneficial to provide additional details about ST screening, particularly the technical advantages of 3D topography and how it differs from traditional clinical screening methods. Expanding on these points would help clarify the central role of ST screening in this study. Additionally, briefly highlighting the unique features of the BackSCNR® system could further emphasize its relevance and innovation.”
We really appreciate your comments on the introduction. We have further explained the potential advantages of ST compared to traditional screening in lines 94-96. Also, a new reference has been accordingly added. Then, we have explained some of the unique features of the software in lines 105-110. We have removed the explanation of the software from methods, and introduce and augmented the detail on it in the introduction.
The presentation of Table 2 could be simplified to enhance clarity. Consider converting the data into a bar graph or other visual representation, as this might help readers grasp patterns more intuitively. Alternatively, emphasizing the columns with average values and providing standard deviation data in the footnotes could make the table more concise and reader-friendly.
Thanks for this suggestion, we have replaced the table with a graph representing the most practiced sports. This has an effect on tables and figure numbers, that have been updated and marked in the text.
While the high specificity of the ST screening model is promising, the sensitivity of 36% raises significant concerns regarding its clinical utility. The authors should provide a more detailed discussion on the potential implications of this low sensitivity, such as the risk of false negatives leading to delayed treatment for mild scoliosis cases. Additionally, the manuscript would benefit from an exploration of the factors contributing to the low sensitivity, such as limitations in detecting mild asymmetries or issues related to the measurement posture.
Thank you for this comment, it has been useful to further elaborate our discussion. Regarding the limitations of the low sensitivity we have added lines 297-299 and also suggest the combination of both screening methods, because one has too many false positives and the other, as you pointed out, too many false negatuves. Then, we consider that this fragment explain why mild asymmetries may have not been detected: “Thus, it is possible that having a mild scoliosis (10°-15° Cobb) does not necessarily imply having a 3D deformity, or at least one that is more relevant than in the non-scoliotic population. Further training of the CNN model with mild cases may increase its accuracy detecting AIS.”
We also consider that these paragraphs explain the potential reasons of the differences between methods in capturing asymmetry and screening scoliosis, but we would appreciate more specific comments if you consider they could be relevant:
“We believe that this significant difference between the screening results is due to the assessment posture used. Our CNN screening model was trained with ST images of participants in a standing position, while the clinical screening is performed in a forward bending position. Thus, the shape features captured and considered by each are different and potentially complementary during the screening process. In addition, the clinical screening is based on an angular measurement of the dorsal hump, while the ST model used deviations and depth maps of the entire torso back. In light of this result, the evaluation of both methods in terms of sensitivity, specificity, PPV, NPV and accuracy gained relevance.”
“Among the positive cases detected by ST, there were 3 cases with a positive ADAMS test and 4° of TRA that did not receive a radiological confirmation, but their topographies show clear signs of potential scoliosis. The DCM has been associated with spinal curve asymmetries and has been used to predict curve severity and progression (29,30). These, shown in figure 4 and compared to a confirmed case of moderate scoliosis, have asymmetry patterns compatible with scoliosis due to the presence of humps and depressions. With the correct radiologic conformation of the condition in these participants, the sensitivity of the ST method would be higher than that currently reported in this study.
